# Chitosan and Chitosan Nanoparticles: Parameters Enhancing Antifungal Activity

**DOI:** 10.3390/molecules28072996

**Published:** 2023-03-27

**Authors:** Pawel Poznanski, Amir Hameed, Waclaw Orczyk

**Affiliations:** Plant Breeding and Acclimatization Institute—National Research Institute, Radzikow, 05-870 Blonie, Poland

**Keywords:** chitin, deacetylation, dispersity, fungi, pathogen, phospholipids, physicochemical, zeta potential

## Abstract

Chitosan (CS), a biopolymer derived from chitin, is known for strong antifungal activity while being biodegradable, biocompatible, and non-toxic. Because of its characteristic it has been widely used in control of fungal pathogens. Antifungal activity of chitosan can be further enhanced by obtaining chitosan nanoparticles (CSNPs). However, most of the experiments using CS and CSNPs as antifungal agents were performed under various conditions and using diverse CS batches of different characteristics and obtained from different sources. Therefore, it is essential to systematize the available information. This work contains a current review on how the CS parameters: molecular weight, degree of deacetylation, acetylation pattern and dispersity of these features shape its antifungal activity. It also considers how concentration and protonation (pH) of CS water solutions define final biological effect. Review explains in detail how CS parameters affect characteristics of CSNPs, particle size, zeta potential, and dispersities of both and determine antifungal activity. In addition to the parameters of CS and CSNPs, the review also discusses the possible characteristics of fungal cells that determine their susceptibility to the substances. The response of fungi to CS and CSNPs varies according to different fungal species and their stages of development. The precise knowledge of how CS and CSNP parameters affect specific fungal pathogens will help design and optimize environmentally friendly plant protection strategies against fungi.

## 1. Introduction

Chitin, the main component of arthropod and insect exoskeleton, present also in fungal cell walls, is one of the most abundant biopolymers on Earth. It consists of N-acetylglucosamine residues linked by a β 1–4 glycosidic bond. Chitosan (CS) is the product of partial deacetylation of chitin and is composed of acetylglucosamine monomers (GlcNAc) and glucosamine monomers (GlcN) (Figure 1) [1]. The deacetylation process, inevitably accompanied by partial hydrolysis, gives a wide variety of CS molecules. The combination of various molecules might be unique for each batch of CS. This great diversity, enlarged by various sources of chitin, is a challenge for CS standardization and comparison of results obtained by different research groups. Commercially available products are characterized by source of the chitin, which might be crustacean of fungal, the type of deacetylation and hydrolysis processes, the degree of deacetylation (DD) expressed as the percentage of GlcN residues, the pattern of acetylation (PA), and the molecular weight (MW).

Chitosan has a strong antifungal effect due to its unique physicochemical properties, biodegradability, and biocompatibility [1,2,3]. Because of its biological activity, chitosan has been used to directly inhibit the growth of several fungal pathogens on crop plants [4,5,6,7,8] as well as indirectly stimulating plant defense mechanisms [9,10]. Furthermore, chitosan has been shown to inhibit the growth of disease-causing human fungal pathogens [11,12,13]. Additionally, good fiber and film formation properties of chitosan made it a suitable for food preservation, edible coating, and packaging [14]. Chitosan and chitosan-based particles can be also used as stabilizing component of Pickering emulsions with wide applications in medicine, cosmetics and food industry [15]. Chitosan was registered by the United States Environmental Protection Agency [16] as a sustainable and safe material for environmental applications [17,18,19,20,21,22,23].

Currently we know that the antifungal activity is shaped by the combination of several attributes such as the original source of chitin (crustaceans or fungal), an average DD and MW, as well as the diversity range of these parameters of a particular batch of CS (Figure 2) [24,25,26,27]. In certain cases, antifungal activity of CS might even exceed that of a commercial fungicide [28]. Diverse species of fungi respond differently to CS applications, and the sensitivity varies depending on their life cycle phase [29,30]. Several mechanisms have been proposed to explain the antifungal activity of CS. They include local disintegration of fungal cell membranes, leakage of cytoplasm, chelation of crucial nutrients, and binding of nucleic acids altering the flow of genetic information [4,31]. As stated above, the process of CS production determines the unique physicochemical properties of the CS batch, which have a huge impact on antifungal activity. The effect of pathogenic fungi by CS, although studied by a number of research groups, is not based on clear guidelines to standardize the experiments. Because of this, the biological material, experimental conditions, and characteristics of the CS samples varied significantly [13,32,33].

As stated by Cord-Landwehr, et al. [34], the chitosan developmental process can be categorized into three generations: (1) first generation chitosan, with poorly defined parameters, varying between subsequent batches (2) second generation chitosan, mostly obtained by chemical methods, with defined DD and MW and very high repeatability of these parameters (3) third generation chitosan which could be called “chitosan of the future” that would be designed with defined values of all parameters: DD, MW, PA and their dispersities [34]. Systematized knowledge about parameters and their relationship with biological effects of chitosan are necessary to make progress for the third generation of chitosan.

The best way to improve the antifungal activity of chitosan is to modify those parameters, which have the strongest effect on these characteristics. As reported in many articles, chitosan-derived nanoparticles (CSNP) show stronger antifungal effect compared to conventional chitosan. Most probably due to the higher surface-to-charge density, bigger surface area, and better cellular uptake compared to bulk chitosan [35,36,37,38]. Different methods of CSNPs preparation such as ionic gelation, ionic crosslinking, covalent crosslinking, precipitation, polymerization, self-assembly, and spray drying have been proposed and optimized [39]. It should be noted that relatively small modifications at any stage can change the parameters and biological properties of CSNPs [40]. The objective of this review is to organize current knowledge on the physicochemical properties of CS with a focus on those that are crucial for its antifungal activity. This may help to design CS-based approaches of protection against pathogenic fungi, considering the inherent antifungal properties of CS variants that might be specific to diverse fungal species. A comprehensive understanding of the mechanisms shaping the antifungal properties of CS and CSNPs may allow for the development of effective alternatives to conventional fungicides that are safe for humans and the environment.

## 2. Determinants of Chitosan Antifungal Activity

### 2.1. Fungi Cell Membrane Composition and Chitosan Susceptibility

Fungi species can be divided into two groups: chitosan susceptible or sensitive and chitosan resistant. Most species of plant pathogenic fungi are sensitive to chitosan, while nematophagous fungi and insect entomopathogens are chitosan resistant [6].

Cell membrane of sensitive species is permeabilized by chitosan [41] and in response to this treatment a set of genes encoding cell wall-related proteins, oxidoreductases, and transport-related proteins are up-regulated [42,43]. It was proposed that negatively charged phospholipids of the plasma membrane were the main target of positively charged chitosan molecules [44] and the notion was confirmed by high ratio of phospholipids in cell membranes of species sensitive to chitosan [45]. In contrary to the above, a detailed analysis of the lipid composition in four phylogenetically distant species showed that the relative content of phospholipids in chitosan susceptible (*Neurospora crassa* and *Fusarium oxysporum*) and chitosan tolerant species (*Pochonia chlamydosporia* and *Beauveria bassiana*) were found to be very similar within each group. The main difference between the two groups was much higher content of polyunsaturated fatty acids (mainly linoleic) and lower content of saturated palmitic, stearic, and monounsaturated oleic acids in sensitive species comparing to the tolerant ones [41].

In line with this was characteristics of *N. crassa* mutant with inactive fatty acid desaturase, and thus devoid of polyunsaturated fatty acids, which showed elevated chitosan tolerance compared to the wild type. The feature, observed in germinating conidia and growing mycelium, indicated that the fluidity of the fungal cell membranes determines the sensitivity to chitosan. The higher the content of polyunsaturated fatty acids and the greater fluidity of the membranes, the greater susceptibility to chitosan [41]. This notion was experimentally confirmed by Zakrzewska, et al. [28]. The authors elevated plasma membrane fluidity in yeast *Saccharomyces cerevisiae* cells by higher growth temperature or miconazole treatment (an inhibitor of ergosterol biosynthesis) and observed a significant increase of yeast sensitivity to chitosan [28]. There are clues that chitin content in fungal cell wall might be important. When it is bigger than 10%, as in chitosan tolerant *Aspergillus niger*, it may be associated with a bigger tolerance to this polymer [46,47].

### 2.2. Fungus Developmental Stages and Chitosan Susceptibility

Restriction of fungal growth by chitosan is not only species-specific but also depends on the fungus developmental stage. In vitro study of *Penicillium expansum* and *Botrytis cinerea* treated with chitosan (15 cps, DD 90%) showed that spore germination was restricted stronger in *P. expansum* compared to *B. cinerea* while mycelial growth of *P. expansum* was less restricted than that of *B. cinerea* [48]. The higher susceptibility of *P. expansum* spores was further analyzed by testing integrity of the spore cell membranes subjected to CS treatment. The authors found lower integrity of *P. expansum* spore cell membranes compared to those of *B. cinerea*, confirming spore germination results. At the same time, the mycelium of *B. cinerea* was more susceptible to CS compared to *P. expansum.* This directly indicates that the sensitivity to chitosan might change during development [48].

In line with the above, Palma-Guerrero, et al. [49] found that cells representing different developmental stages of *Neurospora crassa* varied in their sensitivity to chitosan. The conidia treated with chitosan sample (MW 70 kDa, DD 79.6%) 100 ppm concentration were killed within less than 4 min, the conidial germlings within 35–45 min, and the vegetative hyphae within 40 min. CS permeabilized the fungal plasma membrane and was detected inside the conidia five minutes after the treatment. The process, associated with rapid Ca^2+^ cellular uptake, destabilized the Ca^2+^ homeostasis and led to the cell death. The authors also found that the process was ATP-dependent, and it did not involve endocytosis [49].

### 2.3. Physiochemical Attributes of Chitosan and Its Antifungal Activity

The results of numerous reported experiments indicate that the CS antifungal activity depends on its physicochemical parameters, however, the lack of standardized experimental conditions does not allow for a direct comparison of different CS batches. Depending on the provider, CS batches are characterized by an average MW in kDa units, by degree of polymerization (DP) expressed as number of monomers in a single polymer molecule and are characterized by viscosity of the standardized water solutions expressed in cps units. Considering these parameters, the CS batches can be roughly divided into the four groups: (1) the batches of very low MW, sometimes designated as CS oligomers or CS oligosaccharides, those could be additionally characterized by the values of their DP, (2) the bathes of low MW (LMW) with an average molecular weight smaller than 100 kDa, (3) the batches with MW ranging from 100 to 1000 kDa and representing medium molecular weight (MMW) chitosan, and (4) the batches of high MW (HMW) with molecules of over 1000 kDa. The CSs, tested by different teams to assess their antifungal activity, represented a broad spectrum of batches with different values of MW, DP or viscosity. Since they represented independently designed experiments with diverse fungi species and a wide range of tested CS samples, the results could be used to draw only general conclusions on CS antifungal characteristics.

Park, et al. [33] determined the Minimal Inhibitory Concentration (MIC) for nine species of fungi (*Candida albicans*, *Trichosporon beigelii*, *Saccharomyces cerevisiae*, *Aspergillus fumigatus*, *A. parasiticus, Botrytis cinerea, Fusarium solani*, *F. oxysporum*, *Penicillium verrucosum*) using chitosan oligosaccharides 1, 3, 5 and 10 kDa. They found that MIC of CSs 5 kDa and 10 kDa did not exceed 0.04 mg/mL for all tested species while MIC of the remaining CSs was bigger. Thus, out of the tested four chitosan batches, the CS 5 and 10 kDa showed stronger antifungal activity than 1 and 3 kDa [33].

Rahman, et al. [50] used chitosan DP 206 (which is as equivalent of 33.4 kDa) as a substrate to generate three types of CS: (1) the sample with DP ranging from 3 to 10 (0.5 to 1.6 kDa), DP 23 (3.7 kDa), and DP 40 (6.5 kDa). The highest antifungal activity against two pathogenic fungi *B. cinerea* and *Mucor piriformis*, based on conidial germination and hyphal growth, was observed for samples DP 23 and DP 40. The activity of the original CS DP 206 and the sample DP 3–10 showed weaker activity than DP 23 and DP 40. The results indicated that the highest antifungal activity was associated with a specific range of MW [50]. Interestingly, one of the pathogens, *B. cinerea*, used in this study, was also tested by Park, et al. [33] and in both cases the most effective range of MW was similar, 6.5 kDa in Rahman, et al. [50] and 5–10 kDa in Park, et al. [33,50].

Li, et al. [26] used chitosan of HMW 658 kDa with a degree of deacetylation of 82% as a substrate to obtain LMW samples: 5.5 kDa, 9 kDa, 18.8 kDa and 41.2 kDa. Based on the inhibition of mycelium growth of the three fungal pathogens *Phomopsis asparagi*, *Fusarium oxysporum,* and *Stemphylium solani* all LMW samples, used in two concentrations 200 and 400 mg/mL, had higher antifungal activity than the original HMW CS. In this group, the strongest antifungal activity was found for CS 18.8 kDa and 41.2 kDa. The intermediate effect was observed for 5 kDa and 9 kDa samples, and the weakest activity was found for the original HMW CS 658 kDa [26]. Compatible results, indicating that the most effective MW range for *F. oxysporum* was 5 and 10 kDa [33]. In case of *Fusarium graminearum*, Luan, et al. [51] have tested four CS with increasing MW of 5.1 kDa, 30.2 kDa, 102.8 kDa and 189.3 kDa (DD = 90%). The highest antifungal activity has found when 102.8 kDa CS was used. This data shows that while low MW CS tend to have weaker antifungal effect, an increase of MW doesn’t always correlate with stronger inhibition of fungal growth [51,52].

The broad spectrum of CS batches with different MW were analysed by Li, et al. [53]. They tested antifungal activity against *Aspergillus niger* using CS samples with 95% DD and MW 50, 140, 200, 800 and 1000 kDa. The highest antifungal effect was observed for the 50 kDa sample. The effect was weaker for 140 kDa and the weakest for 200 kDa. Surprisingly, the CS samples 800 kDa and 1000 kDa stimulated the growth of the fungus. Different chitosan samples not only affected the growth of *A. niger*, but also changed morphology and ultrastructure of the hyphae, which were detected under transmission electron microscope (TEM). The 1000 kDa CS sample, which promoted fungal growth, did not show any impact on the morphology of the hyphae, while the surface of the fungal cells treated with CS 50 kDa, the one with the strongest antifungal activity, was irregular and visibly damaged. Consistent with the above, were diverse membrane penetration rates by CS samples of different MW. The process was tracked using 50 kDa and 1000 kDa FITC-labeled chitosan samples. The 50 kDa sample easily penetrated the plasma membrane of *A. niger*, while the CS 1000 kDa remained outside the fungus cell [53]. The authors suggested that the stronger antifungal activity of lower MW samples could be explained by the smaller polymer molecules and the relatively bigger number of free amine groups that interacted with fungal membranes [53].

### 2.4. Chitosan Degree of Deacetylation and Antifungal Activity

The degree of deacetylation (DD) is another parameter of CS essential for its antifungal activity. Younes, et al. [13] investigated the antifungal activity of chitosan with varying MW from 42.5 to 135 kDa and DD from 39% to 98% testing three species: *A. niger*, *F. oxysporum*, and *A. solani*. All were affected by the tested chitosan samples, but the response pattern was strongly species dependent. Growth inhibition of *A. solani* and *F. oxysporum* clearly depended on chitosan DD, indicating that higher DD values had stronger antifungal activity. The inhibitory effect on *F. oxysporum* depended mainly on MW, but at the same time, DD greater than 59% was required to restrict growth. The growth of A. niger was restricted by chitosan; however, no effect of different DD and MW was observed [13].

Tsai, et al. [27] compared the antifungal effect of CS on two groups of species: susceptible (*Candida albicans, F. oxysporum*) and resistant (*Aspergillus fumigatus*, *Aspergillus parasiticus*). The tested CS samples had DD 56% to 98% and MW from 51 to 1080 kDa. The results showed that the strong antifungal effect, assessed by Minimum Lethal Concertation (MLC), depended on the DD of the chitosan sample. As expected, it was observed only for susceptible species while the growth of tolerant species was not restricted [27].

Diverse studies generally confirmed a positive correlation between antifungal activity and DD of the CS sample. It was explained by strong electrostatic interaction between fungal cell membranes and CS amine groups which were protonated in solutions with pH value lower than protonation constant p*K*_a_ of the particular chitosan sample. The relative number of the groups was directly associated with the DD of the CS molecules. The effect caused by deacetylation may overlap, to some extent, with the effect of the MW (Figure 3). According to several groups, CS molecules of high MW may form internal hydrogen bonds, which lower the effective number of amine groups capable for interaction with the membranes [13,27,53].

### 2.5. Chitosan Pattern of Acetylation and Antifungal Activity

Although DD informs about the average rate of deacetylated monomers of the entire CS sample, the pattern of acetylation (PA) depicts the sequence of acetylated monomers on a single chitosan particle. PA is the parameter that has not been thoroughly investigated till now, but with recent advancements of analytical methods the gap is expected to be filled [56,57,58]. Chitosan samples with a specified PA is hardly commercially available. Most commercially available samples are obtained by a chemical process of heterogeneous or homogeneous deacetylation of chitin. CS samples obtained through either of these processes have random PA, which is different in different polymer molecules present in one sample. In other words, the methods lead to uncontrolled differences between different CS batches. Currently, it is not possible to test the effect of PA unless highly specialized methods of obtaining and characterization of CS with specific PA are employed. In consequence, there is still a small amount of data on how PA affects the biological activity of CS. Recently developed methods of enzymatic deacetylation of chitin open new possibilities to obtain chitosans with specified, nonrandom PA [59]. One of the first article in this field, by Sreekumar, et al. [59], shown that enzymatically obtained chitosan with block-PA (DD 67%, DP 800) had different physicochemical characteristics compared to chemically obtained chitosan with random-PA (DD 66%, DP 700). They found that, water solution of block-PA chitosan had lower viscosity than that of random-PA chitosan. Also, block-PA chitosan showed stringer hydrophobicity what, according to the authors, was the result of bigger proportion of hydrophobic block domains compared to the random-PA chitosan samples. Although they did not test directly antifungal activity of samples with different PA types, they found that block-PA chitosan samples exhibited stronger antimicrobial activity against *Pseudomonas syringae* pv. *Tomato*. Testing different CS samples with block-PA, the authors reported stronger antibacterial activity of samples with bigger DD values. As they discussed, this was the result of higher charge density of the molecules, stronger electrostatic interactions between GlcN-rich blocks with membrane components, and, in consequence, more evident disruption of cell integrity [59]. Similar effect concerning antifungal activities of CSs with random PA, reported in many articles, was discussed earlier.

Additionally, as it was shown by Basa, et al. [58], different PA of chitosan oligomers activated different levels of plant priming. This observation indicated that defined PA could be recognized by plant cells and could activate distinct plant response. This set of results might also indicate that the biological activity of chitosan with different Pas comes not only from the electrostatic interactions, but also from the specific recognition of chitosan molecules with defined PA by plant receptors which in turn would trigger specific signaling pathway. More tests are required, but this information could open a new chapter in designing CS with specific PA to maximize its antifungal activity.

### 2.6. Dispersity of the Chitosan Sample Promotes Antifungal Activity

Most of the CS samples are heterogeneous, meaning that they consist of various fractions with different molecular weight/degrees of polymerization (MW/DP), deacetylation degrees (DD), and different patterns of acetylation (PA). For each of the basic CS parameters (i.e., MW/DP, DD and PA) the dispersity parameter (Ð) was introduced. It provides farther information on the variety of fractions present in a defined CS sample. Ð_MW_/Ð_DP_ indicates the variety of size fractions present in a single chitosan sample. Ð_DD_ defines the dispersity of chitosan polymers with different DD. Ð_PA_ describes how differently acetylated monomers are distributed in the polymer molecule (or how many molecules with a specific sequence are present in the specified chitosan sample) [58]. So far, only the antifungal effect of CS dispersity of MW/DP (Ð_MW_/Ð_DP_) was investigated.

Attjioui, et al. [32] tested inhibition of *F. graminearum* growth using different CS samples; the sample with DP 300, Ð_DP_ 1.24, the product of its hydrolysis with the average DP 70 and the two fractions of the hydrolysate: DP 90 and oligomers DP 2 to 17. All samples had the same DD 90%. Both, the initial chitosan sample (DP 300) and the product of its hydrolysis (DP 70) had a similar MIC at 100 µg/mL. Tested separately, the larger polymer fraction (DP 90) had similar antifungal activity as the parent chitosan, whereas the oligomer fraction (DP 2–17) had much weaker activity with MIC 200 µg/mL. The results implied a synergy of antifungal activities of both fractions when applied together (Figure 4). According to the authors, the synergy was the result of destabilization of the cell membranes by long CS polymers, which allowed the CS oligomers to penetrate the fungal cell and to interact with intracellular components. The authors concluded that oligomers showed very weak antifungal activity because molecules of this fraction could not disrupt fungal membrane. As reported previously, disruption of the membranes by fraction of longer polymers was required for efficient reduction of fungi growth [32].

Lemke, et al. [60] confirmed observations that fractions of chitosan differing in size act in synergy in decreasing fungal growth. Antifungal activity was evaluated by measuring CS inhibitory concentration (IC50), which restricted *F. graminearum* growth by 50%. They found that the IC50 of the polymeric CS fraction (MW 43.3 kDa, DP 250) was 155 µg/mL, the IC50 of the oligomeric fraction (MW 2–4 kDa. DP 2–15) was 739 µg/mL while a mixture of the two had IC50 at 133 µg/mL. However, it should be noted that the antifungal activity of the original CS (MW 58.7 kDa, DP 347) was much greater with IC50 at 25 µg/mL [60].

### 2.7. Effect of pH on Chitosan Antifungal Activity

CS sample can be dissolved in an acidic solution when the pH value is lower than the protonation constants (p*K*_a_) of this sample. The p*K*_a_ of the CS is narrowly increasing from 6.39 to 6.51 when the MW changes from 60 to 1370 kDa. A similar and narrow range of p*K*_a_ changes from 6.51 to 6.17 is associated with DD from 73.3 to 94.6% [61]. The p*K*_a_ is determined by the MW and DD of the CS sample, and these parameters, plus the solubility in water and the antifungal activity of the CS sample, are interconnected.

Alburquenque, et al. [62] treated 30 strains of *Candida* spp. With CS (70 kDa, DD 75%) and found that for 69% strains the minimal inhibitory concentration at pH 4.0 was 4.8 mg/L while at pH 7.0 it ranged depending on strain and the values of MIC were from 1 to 4-times bigger. At low pH, the CS amine groups are protonated, and the CS molecules become polycationic. Biological activity is the direct result of the positive charge of protonated amine groups which are responsible for interaction of protonated chitosan molecules with negatively charged proteins, fatty acids, lipids, and nucleic acids of the fungal cell. This, in turn, leads to disintegration of fungal cell membranes, sequestration of cell components, and observed antifungal activity [13,27,53].

## 3. Chitosan Nanoparticles: The Way to Increase Antifungal Activity

CS molecules in mildly acidic solutions can form stable colloidal particles (Figure 5). These self-assembled aggregates were detected in solutions of CS polymers and in solute ions of CS oligomers [63,64,65].

The formation of the aggregates depends on the hydrophobic interactions and hydrogen bonding [63]. The CS aggregates can be further stabilized using different methods leading to formation of CS nanoparticles (CSNPs). These particles combine the biological activity of chitosan and the properties specific to nanoparticles. Several methods of CSNPs formation have been described. The list includes ionic gelation [66,67], emulsion crosslinking [68], spray drying [69], emulsion-droplet coalescence [70] and reverse micellar method [71]. The method of ionic gelation is technically simple and is the most frequently used to obtain CSNPs to control fungal pathogens. In this approach, the self-aggregates of protonated CS molecules are stabilized by cross-linking with negatively charged sodium tripolyphosphate (TPP) (Figure 5). The resultant particles are stable. Comparing to CS self-aggregates they are more compact, and with a higher charge density [72,73]. Two main parameters of CSNPs, that is particle size and zeta potential, can be further adjusted to obtain a homogeneous suspension with desired characteristics [74,75]. As reported in many articles, relatively small modifications at any stage of nanoparticle preparation, as well as the characteristics of the original CS sample (MW, DD, PA) affect the parameters and biological properties of obtained nanoparticles [40,59,73]. Depending on the physicochemical properties of the initial chitosan sample and the method of nanoparticle formation, their size ranged from 40 to 600 nm and the zeta potential was from 16 to 54 mV. The size of CSNPs can be modulated by the pH of the suspension. The more acidic the pH of the suspension, the larger CSNPs are in size. This change in size can be reversed. As presented by Zhu, et al. [76] changing between pH 6.3 and pH 4 allowed a respective decrease and increase in size of CSNPs [76]

As reported in many articles the chitosan-derived nanoparticles showed a stronger antifungal effect compared to bulk chitosan. This was probably due to the higher surface-to-charge density, bigger surface area, and better cellular uptake [35,36,37,38,77]. However, in some cases, the bulk CS showed higher antifungal activity [4]. Ing, et al. [35] tested the antifungal activity of CSNPs prepared from LMW (70 kDa, DD 75–85%) and HMW (310 kDa, DD 85%) chitosan against *C. albicans*, *F. solani* and *Aspergillus niger*. They found that the MICs of NPs obtained from LMW and HMW CS tested against *C. albicans* and *F. solani* were 0.3–1.2 mg/mL and 0.6–1.2 mg/mL, respectively. The MIC of both types of CS was 3 mg/mL, indicating that the activity of the particular CS sample was 3 to 10 times lower than activity of obtained from this sample nanoparticles. It is worth noting that the growth of *A. niger*, which was not affected by neither CS type, was inhibited only by NPs obtained from HMW chitosan [35].The sizes of NPs formed in solution of LMW CS were from 174 to 255 nm and zeta potential from 39 to 48 mV when concentrations of CS was from 1 to 3 mg/mL. The size diversity of NPs formed from a single concentration of HMW CS was from 201 to 301 nm, while zeta potential ranged from 40 to 54 mV. From this set of NPs, only one class, 300 nm in size and 54 mV of zeta potential was found to inhibit the growth of the most tolerant *A. niger* [35].

Saharan, et al. [78] found that the growth of *Alternaria alternata*, *Macrophomina phaseolina* and *Rhizoctonia solani* was inhibited by 82%, 88%, and 34% in the presence of 0.1% CSNP, while inhibition was only 21%, 18% and 17% in the presence of the same concentration of CS (MW not specified, DD = 80%). Similar rates were observed for *A. alternata* spore germination. In the presence of 0.1% CSNPs the 87% of the spores did not germinate while only 21% of the spores were inhibited by 0.1% of the CS. The growth of all tested species was more affected by CSNPs than CS [78]. The results also confirm, as discussed earlier, the different sensitivity of fungal species to treatment with CS and CSNPs. The most effective concentration was 0.1% (1000 ppm). It was much higher compared to other reports, possibly because the authors did not preselect the most effective CS type.

When discussing the results, it should be noted that fungal growth is also retarded by low concentrations (in the range of 0.001–0.01% range) of acetic acid. Although this component should be included in the mock treatment for the proper reference, this factor was rarely considered in the reports.

### Particle Size and Zeta Potential Define Antifungal Activity of Chitosan Nanoparticles

The antifungal activity of CSNPs depends on the physicochemical parameters of the parental chitosan (MW/DP, DD, PA) as well as the conditions of the nanoparticle formation (Table 1). The levels of antifungal activities of CSNPs differed for various species of fungi [35,79] tested response of three fungal species, *C. albicans*, *F. solani*, and *A. niger* using wide range of different CSNPs obtained from low and high molecular weight CS samples. The NPs obtained from LMW CS had average sizes of 174, 233, and 255 nm, and zeta potential 39, 38, and 48 mV respectively. The NPs obtained from HMW CS had average sizes of 210, 263 and 301 nm and zeta potential 40 mV, 52 mV and 54 mV respectively. The strongest antifungal effects on *C. albicans* were found when CSNPs obtained from LMW CS had the average size 174 nm and zeta potential l39 mv. CSNPs made from HMW CS showed a different pattern of activity. The strongest antifungal activity was found when NPs had the biggest average size (301 nm) and the biggest zeta potential (54 mV). According to the authors, NPs obtained from LMW CS compared to NPs from HMW CS, had stronger impact on intracellular processes because they easier entered the cells due to their small size and low zeta potential.

Another pattern of biological activity found for NPs from HMW CS. The observed impact on cell membranes was more destructive due to bigger size (bigger cumulative area of the particles) and bigger values of zeta potential. The growth of *F. solani* and *A. niger* was inhibited only by the biggest NPs (263 and 301 nm) which were generated from HMW CS. This type of NPs that is big average size and low zeta potential, showed stronger antifungal activity compared to NPs of the opposite parameters [35].

## 4. Conclusions and Prospects

The chitosan sample can be characterized by several physicochemical parameters such as molecular weight (MW), degree of deacetylation (DD), and pattern of acetylation (PA). Each of these parameters is detail in further defined by its dispersity, which indi-cates the variation of the parameter within a chitosan sample. The antifungal activity of a particular chitosan sample is strongly dependent on these parameters, including their dispersity. This biological effect also depends on the concentration of chitosan and the protonation values of the solution (pH). Because of this, antifungal activity could significantly vary between different batches of commercially available chitosan. Furthermore, the observed antifungal activity of a particular batch of chitosan is strongly species-dependent, indicating that the reactions shown by different species of fungi range from sensitive to tolerant.

Chitosan nanoparticles obtained from a particular chitosan batch of chitosan (mostly by ionic gelation method) represent another form of this biopolymer. They are further characterized by the average size [nm], the zeta potential [mV] and the dispersity of both. As reported in many articles, chitosan nanoparticles are more antifungally potent than the original chitosan sample. This feature is clearly associated with the surface potential, size, and dispersity of the nanoparticles. These parameters depend on the original chitosan sample and can be further fine-tuned by the conditions of nanoparticle synthesis.

Chitosan is a biopolymer with versatile biological activity, with one set of parameters, it can act as a strong fungicide, while with different characteristics, it can stimulate the growth of fungi. Furthermore, chitosan can be modified into chitosan nanoparticles with even stronger antifungal activity and better stability in the environment. The range of biological activities of different CS and CSNPs samples result from the interaction of positively charged amino groups of chitosan and negatively charged molecules of fungus cells. The process depends on the parameters of the original chitosan sample, as well as the zeta potential, and the size of the nanoparticles. Understanding how these parameters affect antifungal activity is a prerequisite for improving it in the generation of CS-based preparations. When progress in the development of chitosan is considered, as in the second generation of chitosan, it is possible to characterize most of the chitosan parameters and to obtaining subsequent batches with very low variability. It should be emphasized that current research of antifungal activity of chitosan builds the foundations for the third-generation molecules of this biopolymer. They will be synthesized according to pre-designed parameters and towards the specific applications. The biocompatibility and environmental safety makes the chitosan-based formulation an attractive complementation or replacement for conventional fungicides.

## Figures and Tables

**Figure 1 molecules-28-02996-f001:**
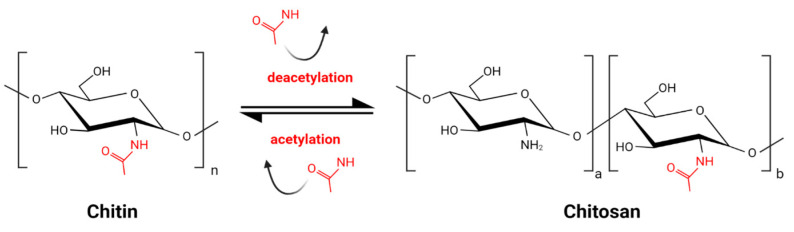
Schematic reaction of partial chitin deacetylation and obtaining chitosan.

**Figure 2 molecules-28-02996-f002:**
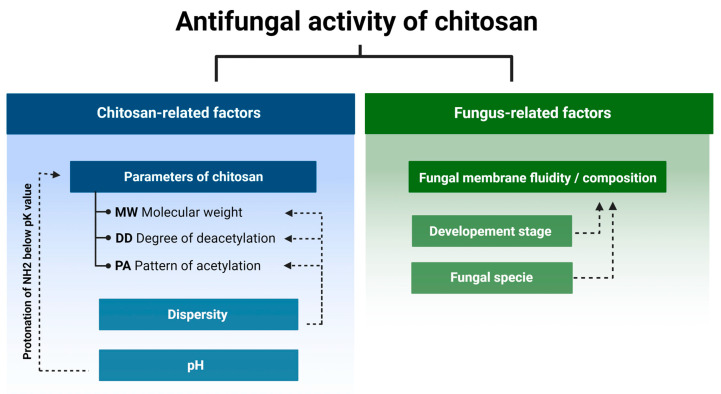
Chitosan and fungus-related factors that affect the antifungal activity of chitosan.

**Figure 3 molecules-28-02996-f003:**
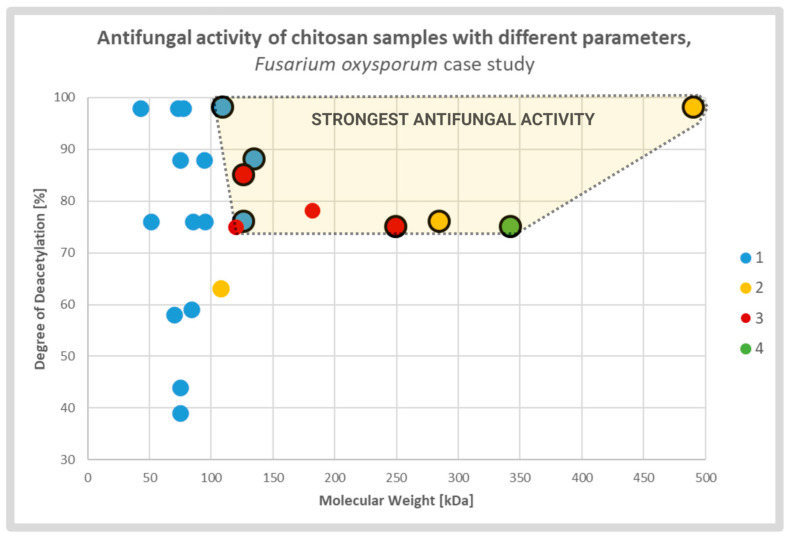
Scatterplot graph of antifungal activities of chitosan samples with different parameters tested on *Fusarium oxysporum*. Chitosan samples showing the strongest antifungal activities are encircled and highlighted. The colors and respective numbers indicate the cited articles: 1 [13] 2 [27] 3 [54] 4 [55].

**Figure 4 molecules-28-02996-f004:**
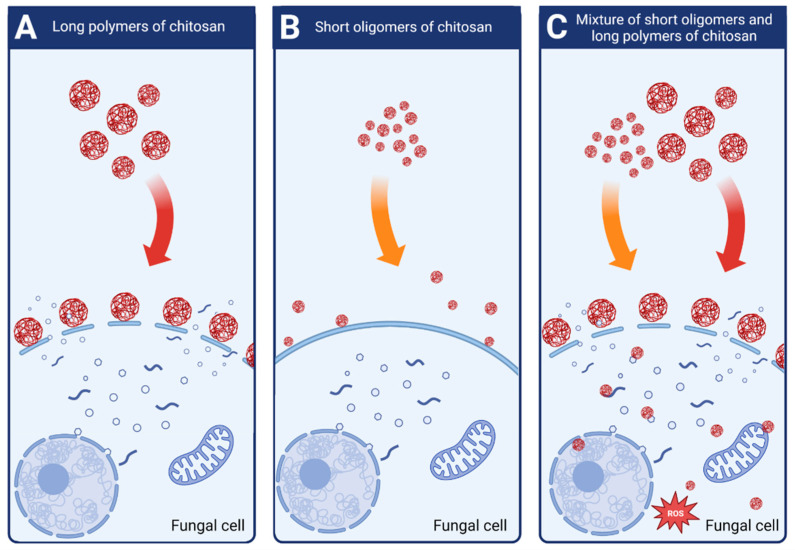
Schematic mechanisms of antifungal activity of different weight fractions of chitosan. (**A**) High molecular weight chitosan destabilizes fungal cell membranes leading to the leakage of intracellular components. (**B**) Low molecular weight chitosan fractions do not destabilize fungal membranes and show very weak antifungal effect. (**C**) Chitosan mixture of high and low molecular weight fractions shows strong antifungal activity. High molecular weight fractions destabilize the membranes allowing the low molecular fractions to penetrate fungal cell and to disturb cell processes.

**Figure 5 molecules-28-02996-f005:**
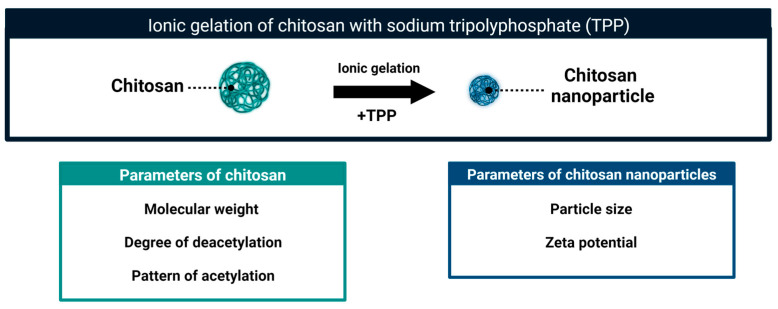
Schematic representation of the formation of chitosan nanoparticles and their basic parameters.

**Table 1 molecules-28-02996-t001:** Parameters of chitosan nanoparticles and their antifungal activity.

Chitosan		Chitosan Nanoparticles					
Molecular Weight [kDa]	Deacetylation [%]	Particle Size [nm]	Zeta Potential [mV]	Fungi Species	Concentration of CSNPs	The Rate of Growth Inhibition	References
70 kDa	75–85%	174 nm	39 mV	*C. albicans*	0.25 [mg/mL]	90%	[35]
				*F. solani*	1.0 [mg/mL]	90%	
				*A. niger*	-	0%	
		233 nm	38 mV	*C. albicans*	0.85 [mg/mL]	90%	
				*F. solani*	0.85 [mg/mL]	90%	
				*A. niger*	-	0%	
		255 nm	48 mV	*C. albicans*	0.60 [mg/mL]	90%	
				*F. solani*	1.21 [mg/mL]	90%	
				*A. niger*	-	0%	
310 kDa	75%	210 nm	40 mV	*C. albicans*	1.0 [mg/mL]	90%	
				*F. solani*	0.5 [mg/mL]	90%	
				*A. niger*	-	0%	
		263 nm	52 mV	*C. albicans*	0.85 [mg/mL]	90%	
				*F. solani*	0.85 [mg/mL]	90%	
				*A. niger*	1.71 [mg/mL]	90%	
		301 nm	54 mV	*C. albicans*	0.60 [mg/mL]	90%	
				*F. solani*	0.60 [mg/mL]	90%	
				*A. niger*	2.42 [mg/mL]	90%	
-	85.61%	-	-	*P.steckii*	5 [mg/mL]	100%	[38]
				*A.oryzae*	>5 [mg/mL]	100%	
“Low molecular weight chitosan”		40–70 nm	48 mV	*F. ox f. radicis lycopersici*	0.0125–0.1%	48–100%	[80]
				*F. oxysporum*		50–100%	
				*F. solani*		50–100%	
				*F. semibaticum*		46–100%	
				*A. solani*		46–100%	
				*P. infestance*		40–100%	
				*R. solani*		51–100%	
				*S. rolfsii*		32–100%	
				*S. sclerotinum*		34–100%	
				*B. cinerea*		39–100%	
				*M. phaseolina*		43–100%	
“High molecular weight chitosan”		40–70 nm	48 mV	*F. ox f. radicis lycopersici*	0.0125–0.1%	55–100%	
				*F. oxysporum*		55–100%	
				*F. solani*		58–100%	
				*F. semibaticum*		52–100%	
				*A. solani*		52–100%	
				*P. infestance*		50–100%	
				*R. solani*		55–100%	
				*S. rolfsii*		39–100%	
				*S. sclerotinum*		46–100%	
				*B. cinerea*		43–100%	
				*M. phaseolina*		53–100%	
244 kDa	86.9%	180 nm	-	*A. tenuis*	0.01–0.08%	20–68%	[81]
				*A. niger*		21–63%	
				*A. terreus*		23–75%	
				*B. bassiana*		74–76%	
				*F. graminearum*		43–60%	
				*F. oxysporum*		32–67%	
				*S. rolfsii*		0–37%	
-	80%	192.2 nm	45.33 mV	*A. alternata*	0.001–0.1%	12–82%	[78]
				*M. phaseolina*		62–88%	
				*R. solani*		13–34%	
“Low molecular weight chitosan”		180.9 nm	45.6 mV	*F. graminearum*	0.001–0.5%	27 –75%	[36]
		225.7 nm	33.4 mV			17–78%	
“Medium molecular weight chitosan”		309.9 nm	33.2 mV			7–52%	
		301.5 nm	20.2 mV			7–53%	
“High molecular weight chitosan”		339.4 nm	21.7 mV			7–52%	
		595.7 nm	16 mV			5–51%	
-	-	100–160 nm	-	*P. expansum*	0.2–0.4 [g/L]	32–44%	[82]

## Data Availability

Not applicable.

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
