# Peer review of "Chitosan and Chitosan Nanoparticles: Parameters Enhancing Antifungal Activity"

_molecules, 2023, doi:10.3390/molecules28072996_

Round 1

Reviewer 1 Report

The review comprehensively discusses chitosan's antifungal properties in relation to agricultural applications. However, there are several missing points that should be added to the review:

  1. There are no cited research papers from 2023, and only two from 2022 were referred to in the review. For an up-to-date review, recent studies should be checked and relevant ones should be added.
  2. The future perspective is missing. What kind of development is the author expecting from future works, or what kind of recommendations or expert opinions do the authors have? There should be an additional paragraph related to this point.

Author Response

Dear Reviewer,

Thank you for reviewing the manuscript. We addressed the suggestions and revised the manuscript accordingly. Below, please find the explanations and answers to the suggestions.

Best regards,

Waclaw Orczyk

Reviewer 1

  1. There are no cited research papers from 2023, and only two from 2022 were referred to in the review. For an up-to-date review, recent studies should be checked and relevant ones should be added.

Explanation: In recent years, we observed multiple work considering complexes of chitosan with different chemical compounds. In our review, the main idea was to combine all information referring to chitosan and chitosan nanoparticles in order to emphasize the antifungal potential of the chitosan itself. Practically all publications that addressed the antifungal activity of chitosan and chitosan nanoparticles in the past two years have been considered and thoroughly reviewed. In response to this suggestion, we included several more publications.

  1. The future perspective is missing. What kind of development is the author expecting from future works, or what kind of recommendations or expert opinions do the authors have? There should be an additional paragraph related to this point.

Explanation: Based on this valuable suggestion, the “Concluding remarks” was further edited and changed to “Conclusions and Future Prospects”  where the most important results and conclusions have been condensed in one paragraph.

Reviewer 2 Report

My comments

Thank you very much for allowing me to cooperate with your respected journal in reviewing this interesting work.

The review is dealing with the “Chitosan and Chitosan Nanoparticles: Parameters Enhancing Antifungal Activity”. This work is very important for readers to improve their awareness regarding the application of nanotechnology in antifungal activity.

·         Chitosan and chitosan nanoparticles have been extensively studied for their antifungal properties. Chitosan is a natural biopolymer derived from chitin, a component of crustacean shells. It has been reported to possess potent antifungal activity against a wide range of fungal species, including Candida, Aspergillus, and Fusarium.

·         The antifungal activity of chitosan and chitosan nanoparticles is dependent on several parameters, including the degree of deacetylation, molecular weight, concentration, pH, and temperature.

·         Studies have shown that chitosan with a higher degree of deacetylation and lower molecular weight exhibits stronger antifungal activity. The size of chitosan nanoparticles also plays a critical role in their antifungal activity, with smaller nanoparticles being more effective.

·         Other parameters that can enhance the antifungal activity of chitosan and chitosan nanoparticles include the addition of other antifungal agents, such as essential oils or silver nanoparticles, and the use of specific preparation methods, such as the ionic gelation method.

·         Overall, chitosan and chitosan nanoparticles show great potential as antifungal agents, and further studies are needed to optimize their properties and explore their potential in various applications, including food preservation, agriculture, and medicine.

1.      The title is precise for the core message of this research.

2.      As you know this work is considered one of the important applied research projects and the word (application) is mentioned only in two lines in the introduction part (lines 42 and 48) and I do not see it in the abstract and/or conclusion. Please specify.

3.      The introduction is well-written and covered enough periods from the literature survey. Some related references on microstructures could be added in the part of applications.

·         Abdelfattah I, El-Shamy AM, Chitosan as Potential De-coloring Agent for Synthetic and Textile Industrial Wastewater, Journal of Environmental Accounting and Management, 2022, 10(3): 305–319. DOI:10.5890/JEAM.2022.09.008

·         Abdelfattah I, El‑Saied FA, Almedolab AA, El‑Shamy AM, Biosorption as a Perfect Technique for Purification of Wastewater Contaminated with Ammonia, Applied Biochemistry, and Biotechnology, 2022, 1-30. https://doi.org/10.1007/s12010-021-03794-4

·         Abdelfattah I, Abuarab ME, Mostafa E, El‑Awady MH, Aboelghait KM, El‑Shamy AM, Integrated system for recycling and treatment of hazardous pharmaceutical wastewater, International Journal of Environmental Science and Technology, 2022, 1-10. https://doi.org/10.1007/s13762-022-04269-7

4.      Figures and tables are acceptable.

5.      The pH is mentioned only one time in the introduction part of the manuscript. As you know pH is very important and is covered well in this review. Some references illustrate the effect of pH on the nanocomposites I hope to see it in the revised form of the manuscript.

·         Zohdy KM, El-Sherif RM, El-Shamy AM, Effect of pH Fluctuations on the Biodegradability of Nanocomposite Mg-Alloy in Simulated Bodily Fluids, Chemical Paper, 2022, 2022: 1-21. https://doi.org/10.1007/s11696-022-02544-y

·         Zohdy KM, El-Sherif RM, El-Shamy A. M., Corrosion and Passivation Behaviors of Tin in Aqueous Solutions of Different pH, Journal of Bio- and Tribo-Corrosion, 2021, 7(2), 1-7. https://doi.org/10.1007/s40735-021-00515-6

6.      The word mechanism is not mentioned completely in the manuscript, and I the schematic mechanisms are already mentioned in this review. I think it is very important for readers to know the mode of action of this composite. Please specify the structure effect relationship.

7.      Please update the conclusion part by adding one or more sentences to highlight the benefits of the application of this nanocomposite as antifungal active material.

8.      The references are listed according to the requirements of the journal.

9.      The manuscript contains some minor editing and grammatical mistakes. Please check it before the publication.

My decision is acceptance after a minor revision.

Author Response

Dear Reviewer,

Thank you for reviewing the manuscript. We addressed the suggestions and revised the manuscript accordingly. Below, please find the explanations and answers to the suggestions.

Best regards,

Waclaw Orczyk

Reviewer 2

  1. The title is precise for the core message of this research.
  2. As you know this work is considered one of the important applied research projects and the word (application) is mentioned only in two lines in the introduction part (lines 42 and 48) and I do not see it in the abstract and/or conclusion. Please specify.

Explanation: The work is focused on the functional application of chitosan with one specific goal which is the antifungal activity of this biopolymer. Because of this, we selected only these publications which were focused on this specific field. In response to this suggestions, in the revised version of the manuscript several more citations have been added and discussed.

  1. The introduction is well-written and covered enough periods from the literature survey. Some related references on microstructures could be added in the part of applications.
  • Abdelfattah I, El-Shamy AM, Chitosan as Potential De-coloring Agent for Synthetic and Textile Industrial Wastewater, Journal of Environmental Accounting and Management, 2022, 10(3): 305–319. DOI:10.5890/JEAM.2022.09.008
  • Abdelfattah I, El‑Saied FA, Almedolab AA, El‑Shamy AM, Biosorption as a Perfect Technique for Purification of Wastewater Contaminated with Ammonia, Applied Biochemistry, and Biotechnology, 2022, 1-30. https://doi.org/10.1007/s12010-021-03794-4
  • Abdelfattah I, Abuarab ME, Mostafa E, El‑Awady MH, Aboelghait KM, El‑Shamy AM, Integrated system for recycling and treatment of hazardous pharmaceutical wastewater, International Journal of Environmental Science and Technology, 2022, 1-10. https://doi.org/10.1007/s13762-022-04269-7.

Explanation: The introduction was revised, and some more references related to the application of chitosan were added.

  1. Figures and tables are acceptable.
  2. The pH is mentioned only one time in the introduction part of the manuscript. As you know pH is very important and is covered well in this review. Some references illustrate the effect of pH on the nanocomposites I hope to see it in the revised form of the manuscript.
  • Zohdy KM, El-Sherif RM, El-Shamy AM, Effect of pH Fluctuations on the Biodegradability of Nanocomposite Mg-Alloy in Simulated Bodily Fluids, Chemical Paper, 2022, 2022: 1-21. https://doi.org/10.1007/s11696-022-02544-y
  • Zohdy KM, El-Sherif RM, El-Shamy A. M., Corrosion and Passivation Behaviors of Tin in Aqueous Solutions of Different pH, Journal of Bio- and Tribo-Corrosion, 2021, 7(2), 1-7. https://doi.org/10.1007/s40735-021-00515-6.

Explanation: The term pH was introduced as one of the parameters that affect protonation of chitosan molecules and further it was presented how the level protonation was associated with antifungal activity. In response to your suggestion, the term pH was also introduced and discussed in sections referring nanostructures of chitosan nanoparticles.

  1. The word mechanism is not mentioned completely in the manuscript, and I the schematic mechanisms are already mentioned in this review. I think it is very important for readers to know the mode of action of this composite. Please specify the structure effect relationship.

Explanation: The mechanism by which chitosan reduces fungal growth is not yet fully explained. In the review we have very briefly pointed out what is the currently proposed mode of action (starting from line 56). Furthermore, we also proposed how the nanostructure characteristics is associated with stronger antifungal effect (line 354, 355). 

  1. Please update the conclusion part by adding one or more sentences to highlight the benefits of the application of this nanocomposite as antifungal active material.

Explanation: Based on this valuable suggestion, the paragraph “Concluding Remarks” was further revised and changed into “Conclusions and future prospects”.

  1. The references are listed according to the requirements of the journal.
  2. The manuscript contains some minor editing and grammatical mistakes. Please check it before the publication.

Explanation: The manuscript was checked, and the errors were corrected.

Reviewer 3 Report

The reviewed manuscript concerns the characteristics of chitosan and chitosan nanoparticles in the context of their antifungal activity. The topic of the manuscript is very interesting and up-to-date due to the properties of chitosan related to the inhibition of the development of various fungal pathogens. This property can be useful in the development of various solutions in agriculture related to the fight against plant pathogens. The manuscript in detail describes the properties of chitosan and its nanoparticles as well as the factors affecting these properties. The number of literature data in this area is limited and dispersed over a large period of time, which is why the cited literature contains a lot of publications older than 5 years. The manuscript is well prepared, easy to read, and I found virtually no errors. I recommend publishing this manuscript.

Technical error: in table 1, the formatting of references is inconsistent with the requirements of the journal.

Author Response

Dear Reviewer,

Thank you for reviewing the manuscript. We addressed the suggestions and revised the manuscript accordingly. Below, please find the explanations and answers to the suggestion.

Best regards,

Waclaw Orczyk

Reviewer 3

Comments and Suggestions for Authors

The reviewed manuscript concerns the characteristics of chitosan and chitosan nanoparticles in the context of their antifungal activity. The topic of the manuscript is very interesting and up-to-date due to the properties of chitosan related to the inhibition of the development of various fungal pathogens. This property can be useful in the development of various solutions in agriculture related to the fight against plant pathogens. The manuscript in detail describes the properties of chitosan and its nanoparticles as well as the factors affecting these properties. The number of literature data in this area is limited and dispersed over a large period of time, which is why the cited literature contains a lot of publications older than 5 years. The manuscript is well prepared, easy to read, and I found virtually no errors. I recommend publishing this manuscript.

Technical error: in table 1, the formatting of references is inconsistent with the requirements of the journal.

Explanation: The errors were corrected, the manuscript formatting is now consisted with the journal requirements.

Round 2

Reviewer 1 Report

The manuscript can be accepted in the present form.